# The Yorkshire Kidney Screening Trial (YKST): protocol for a feasibility study of adding non-contrast abdominal CT scanning to screen for kidney cancer and other abdominal pathology within a trial of community-based CT screening for lung cancer

Juliet A Usher-Smith [ID],[1] Angela Godoy,[2] Sarah W Burge,[3] Simon Burbidge,[4] Jon Cartledge,[5] Philip A J Crosbie [ID],[6] Claire Eckert,[7] Fiona Farquhar,[8] David Hammond,[8] Neil Hancock,[9] Gareth R Iball,[10] Michael Kimuli,[5] Golnessa Masson,[1,11] Richard D Neal [ID],[12] Suzanne Rogerson,[8] Sabrina H Rossi,[2] Evis Sala,[13,14] Andrew Smith,[15] Stephen J Sharp,[16] Irene Simmonds,[7] Tom Wallace,[17] Matthew Ward,[7] Matthew E J Callister,[18] Grant D Stewart [ID] [2]

JAU-S and AG contributed equally.

For numbered affiliations see end of article.

**Correspondence to**
Professor Grant D Stewart;
gds35@cam.ac.uk

## ABSTRACT

**Introduction** Kidney cancer (renal cell cancer (RCC)) is the seventh most common cancer in the UK. As RCC is largely curable if detected at an early stage and most patients have no symptoms, there is international interest in evaluating a screening programme for RCC. The Yorkshire Kidney Screening Trial (YKST) will assess the feasibility of adding non-contrast abdominal CT scanning to screen for RCC and other abdominal pathology within the Yorkshire Lung Screening Trial (YLST), a randomised trial of community-based CT screening for lung cancer.

**Methods and analysis** In YLST, ever-smokers aged 55–80 years registered with a general practice in Leeds have been randomised to a Lung Health Check assessment, including a thoracic low-dose CT (LDCT) for those at high risk of lung cancer, or routine care. YLST participants randomised to the Lung Health Check arm who attend for the second round of screening at 2 years without a history of RCC or abdominal CT scan within the previous 6 months will be invited to take part in YKST. We anticipate inviting 4700 participants. Those who consent will have an abdominal CT immediately following their YLST thoracic LDCT. A subset of participants and the healthcare workers involved will be invited to take part in a qualitative interview. Primary objectives are to quantify the uptake of the abdominal CT, assess the acceptability of the combined screening approach and pilot the majority of procedures for a subsequent randomised controlled trial of RCC screening within lung cancer screening.

**Ethics and dissemination** YKST was approved by the North West-Preston Research Ethics Committee (21/NW/0021), and the Health Research Authority on 3 February 2021. Trial results will be disseminated at clinical meetings, in peer-reviewed journals and to policy-makers.

## STRENGTHS AND LIMITATIONS OF THIS STUDY

⇒ Yorkshire Kidney Screening Trial (YKST) is the first study to investigate whether it is acceptable and feasible to combine lung and kidney cancer screening using non-contrast low-dose CT scanning.

⇒ By nesting YKST within an ongoing trial of lung cancer screening and performing the abdominal scan immediately after the lung scan, the additional abdominal screening can be conducted with very little additional cost or inconvenience to participants.

⇒ The participants invited to take part in YKST are those who have already consented to take part in a lung screening trial and are considered at high risk of lung cancer so may not be representative of those invited to future cancer screening.

⇒ A nested substudy will enable assessment of psychological, social and financial harms, and dissatisfaction with healthcare.

Findings will be made available to participants via the study website (www.YKST.org).

**Trial registration numbers** NCT05005195 and ISRCTN18055040.

## INTRODUCTION

Kidney cancer (or renal cell cancer (RCC)) is the seventh most common cancer in the UK, and incidence is increasing.[1] As with other cancers, survival is strongly dependent on stage at diagnosis: 5-year survival is 87% in stage I compared with 12% in stage IV.[1]

Diagnosing RCC at an early stage is therefore central to improving survival.[2] A particular challenge for the diagnosis of RCC is that 60% of patients are asymptomatic, rising to 87% when considering only stage I cancers.[3] As a result, up to a third of patients present with incurable stage IV disease[2] and half of all patients developing the disease die from it.[1]

The fact that RCC incidence is increasing, is largely curable if detected early and most patients are asymptomatic at the time of diagnosis, has resulted in interest from both the scientific community and patient representatives for the development of an RCC screening programme. In particular, screening and early detection of RCC has been identified as a key research priority in three independent priority setting initiatives over the last 5 years.[4–7] However, despite three decades of interest in the topic, no definitive studies have been conducted and there remain a number of key uncertainties.[8] These include whether detecting RCC earlier would translate into reductions in mortality or lead to overdiagnosis and overtreatment and whether the benefits at population level would outweigh the potential physical, psychosocial and financial harms. Randomised controlled trials are, therefore, needed. Additionally, despite the increasing incidence, extrapolating from studies in the USA or Japan, the prevalence of RCC among middle-aged adults within the general population in the UK is estimated to be 0.21% (95% CI, 0.14 to 0.28%).[9] This means that approximately 500 individuals would need to be screened to identify one person with a RCC unless screening was targeted towards higher-risk individuals.[8]

The gold standard test for detecting and investigating renal masses is a contrast-enhanced abdominal CT scan. It is not feasible to use a contrast-enhanced CT as a stand-alone screening test for RCC due to the relatively high radiation dose and cost, particularly given the low prevalence of RCC. However, using a non-contrast CT scan and combining that with the thoracic low-dose CT (LDCT) scans recommended in the USA in adults aged 50–80 years who have a 20 pack-year smoking history or have quit smoking within the past 15 years[10] and currently being reviewed by the UK National Screening Committee has been proposed.[8] Over 95% of deaths from RCC in the UK occur in those aged over 50 and the relative risk for RCC compared with never smokers is 1.35 for current smokers and 1.22 for ex-smokers. This combined approach would therefore reduce both the costs and radiation, while also targeting those at higher risk due to their age and smoking status.

The potential benefits of using CT to detect RCC have been seen in one of the randomised control trials of lung cancer screening in the USA[11] in which participants diagnosed with RCC within 12 months of the thoracic scan who had a reported abnormality in the upper abdomen had a significantly shorter median time to diagnosis that those without an abnormality in the upper abdomen. However, the thoracic LDCT used within lung cancer screening only includes the upper pole of the kidneys. Additionally,

for any screening programme to be successful, eligible individuals need to take up the offer of screening. Previous research has shown that providing combined 'one stop' cancer screening programmes is viewed positively by members of the public[12] and a survey of over 1000 individuals found that 95% would be 'likely' or 'very likely' to take up an abdominal CT for RCC screening if it was offered in addition to lung cancer screening.[13] These studies, however, report only intention, and not actual attendance, as no such screening programme currently exists. There are also no studies piloting the additional logistics required for such a combined screening programme.

The Yorkshire Lung Screening Trial (YLST) is a community-based, lung screening programme that has recruited individuals who are current or ex-smokers, 55–80 years of age and at high risk of developing lung cancer as defined by the $LLP_{v2}$ score,[14] $PLCO_{M2012}$[15] score or using the 2014 USPSTF criteria.[16] Participants are being invited back for a second thoracic LDCT after 2 years. Nested within YLST, the Yorkshire Kidney Screening Trial (YKST) will take advantage of this unique opportunity to assess the feasibility and acceptability of offering an additional non-contrast abdominal CT at the same time as the thoracic LDCT as a combined abdominal and lung cancer screening approach and to estimate other key uncertainties needed to inform a health economic analysis and subsequent randomised controlled trials within future screening programmes.

## OBJECTIVES

The primary study objectives are as follows:

1. To quantify the uptake of non-contrast abdominal CT to screen for RCC and other abdominal pathology as part of a combined screening modality with thoracic LDCT within a lung health check.
2. To assess the acceptability to patients of combined lung and RCC screening by non-contrast CT scanning.
3. To evaluate the logistics and acceptability to healthcare professionals involved in the combined lung and RCC screening pathway.
4. To pilot the majority of procedures for a subsequent full-scale randomised controlled trial of RCC screening by non-contrast CT scanning within lung cancer screening.

The secondary study objectives are to estimate:

1. The prevalence of renal masses and RCC found on non-contrast CT screening in an appropriate age (55–80 years) and risk group (smokers and ex-smokers).
2. The stage distribution of RCC identified through non-contrast CT screening.
3. The prevalence of incidental renal findings on non-contrast CT scanning.
4. The prevalence of non-renal findings on non-contrast CT scanning.
5. The incidence of RCC in the upper pole of the kidney over sequential non-contrast CT scans.

## OUTCOME MEASURES

The primary outcome measures are:

1. The proportion of individuals invited to have an additional abdominal CT while attending a second round of lung cancer screening who take up the offer of the abdominal CT.
2. The acceptability to participants of combined lung and RCC screening by non-contrast CT scanning.
3. The acceptability to healthcare professionals involved in the combined screening approach.
4. The additional time required for the combined screening approach.

The secondary outcome measures are:

1. The proportion of participants found to have a renal mass or RCC to provide an estimate of the prevalence of RCC found on non-contrast CT screening in 55–80 years smokers and ex-smokers.
2. The stage distribution of RCC identified through non-contrast CT screening.
3. The proportion of participants found to have incidental renal findings on non-contrast CT scanning.
4. The proportion of participants with non-renal findings on non-contrast CT scanning
5. The proportion of RCCs found on the upper pole of participants at the second thoracic screening round

who did not have them in the baseline round, to estimate the incidence of RCC over sequential non-contrast CT scans.

Data will also be collected on further investigations, procedures and management of findings identified on abdominal CT to estimate the individual and health system burden of incidental findings and on the agreement of radiologists reporting the scans and the abdominal CT scan dose and quality to assess the safety. We will also collect long-term (10-year) follow-up data on RCC and other abdominal pathology and apply for CAG approval to obtain data on RCC among participants within YLST who were not invited to take part in YKST.

## METHODS AND ANALYSIS

### Study design

YKST is a non-randomised feasibility study of adding an abdominal CT scan to the thoracic LDCT offered to participants 2 years after recruitment into the YLST.[17]

### Participants and recruitment

Participant recruitment is detailed in figure 1. Participants will be recruited from those attending the second (T2) round of screening within YLST from May 2021 to

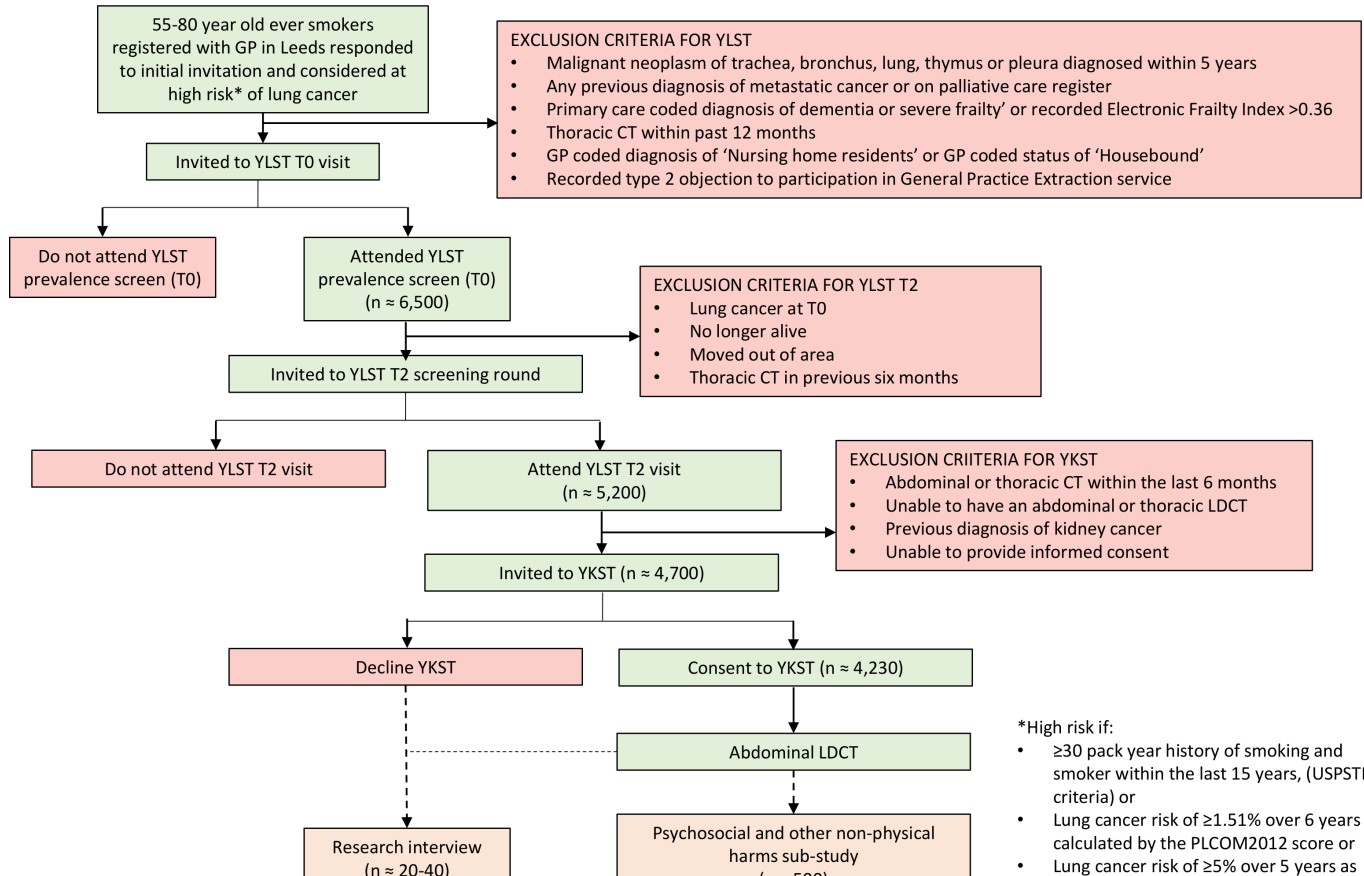

**Figure 1** Study recruitment. GP, general practice; LDCT, low-dose CT; LLP, Liverpool Lung Project; PLCO, Prostate, Lung, Colorectal and Ovarian; USPSTF, US Preventive Services Task Force; YKST, Yorkshire Kidney Screening Trial; YLST, Yorkshire Lung Screening Trial.

October 2022. Full details of YLST are published elsewhere.[17] In brief, YLST is a two-arm (1:1) implementation study using a single-consent Zelen's randomised controlled design with participants randomised to a Lung Health Check or usual care. Participants randomised to the intervention arm are invited to contact a telephone line for a lung cancer risk assessment. Those at high risk of lung cancer are offered a Lung Health Check appointment at a mobile unit sited in convenient community locations, including LDCT screening for lung cancer. The YLST screening programme includes a baseline visit (T0), where participants undergo baseline measurements of height and weight, spirometry (pre-SARS-CoV-2 pandemic), oxygen saturation and exhaled carbon monoxide alongside a smoking cessation intervention and a thoracic LDCT, and a second visit 2 years later (T2), where participants are offered a further thoracic LDCT.

Eligibility for YLST is detailed in figure 1. All those not diagnosed with lung cancer or any other metastatic cancer following the YLST baseline visit (T0) are invited back for T2 between May 2021 and October 2022. The exclusion criteria for YKST at that point are as follows:

▶ Abdominal or thoracic CT within the last 6 months.
▶ Unable to have an abdominal or thoracic LDCT.
▶ Previous diagnosis of RCC.
▶ Unable to provide informed consent.

### Invitation process, consent and baseline data collection

The study processes are shown in figure 2. Participants who attend the mobile van for their YLST T2 visit will be informed of YKST on the van. They will be invited to view the YLST T2 Patient Information video, followed directly by the YKST information video. The YKST video explains the context of the study and the benefits and harms of the additional abdominal CT, including an estimate that in about 5 out of 1000 eligible people the scan may show evidence of RCC, the uncertainty over whether detecting cancers in this way reduces deaths from RCC, the risks associated with the radiation dose and overdiagnosis and the potential to cause anxiety and worry. Participants are also provided with a written participant information sheet covering the same information (online supplemental file 1). A YLST consultation will follow, at the end of which participants will be asked if they would like to take part in YKST. Translation services are offered to patients where required. Eligibility will be checked and fully informed written consent obtained. As part of this consent, participants will consent to allowing the YKST research team access to their medical records.

Participants who consent, as well as those who decline the additional scan, will be invited to take part in a qualitative interview. Participants, who consent to being contacted about an interview, will be asked to provide their contact details. A separate participant information sheet and consent form will be sent to them and they will be asked to contact a qualitative researcher to arrange an interview.

After providing informed consent, participants will complete a short YKST baseline questionnaire asking whether they have a diagnosis of diabetes or hypertension, whether they take antihypertensive medication, if they have a family history of kidney or pancreatic cancer and their average weekly alcohol consumption. Sociodemographic data (age, sex, socioeconomic status and educational level) and height, weight and smoking status will be obtained from data collected in YLST.

Participants will then be shown to a separate room on the van to have the YLST thoracic LDCT, followed immediately by the YKST abdominal CT. To ensure that only those participants who have consented to YKST receive the additional abdominal CT, radiographers will only perform the abdomen scan for those participants who have (1) signed the YKST consent form, and (2) from whom they have received a YKST LDCT request card.

### CT scanning protocol

The scanning protocol for the non-contrast abdominal CT will be based on the protocol used for kidney ureter and bladder scans within Leeds Teaching Hospital Trust (LTHT) and will be reviewed and monitored by the LTHT medical physics team to ensure that the lowest possible dose allowing interpretable images is used for the YKST abdominal images. A 64-channel (or higher) mobile multidetector CT will be used throughout the study. Participants will lie supine on the CT table with arms above their head and thorax and abdomen in the midline of the scanner. Subject comfort will be optimised and maximal inspiration rehearsed prior to the scan to minimise motion during the CT. Imaging will then be performed during suspended maximal inspiration with the standard scanogram used to localise the start and end positions of the scan. No intravenous contrast material will be administered. The kidneys will be scanned in their entirety in a single craniocaudal acquisition and transaxial images of 1 mm thickness will be generated, with further reconstructions as necessary. Radiation exposures will be kept as low as possible while maintaining good image quality. The average CT dose index (CTDIvol) and Dose Length Product for 70–80 kg patients will be monitored to ensure that they closely match the current typical values of 5 mGy and 110 mGycm, respectively, from LTHT scanners. The X-ray tube current (mAs) settings will be automatically varied by the scanner according to participant body habitus. The CT images will then be transferred from the mobile unit to LTH PACS system within 2 days.

### CT scan reporting

A team of LTHT uroradiology consultants will report the abdominal CT scans. They will receive them through the LTHT (PACS) systems and will report them within 2 weeks of the date of the scan. Scans and reports will be stored on the LTHT electronic patient record (PPM+). The time taken to access and generate the report for these scans will be collected as part of a process evaluation.

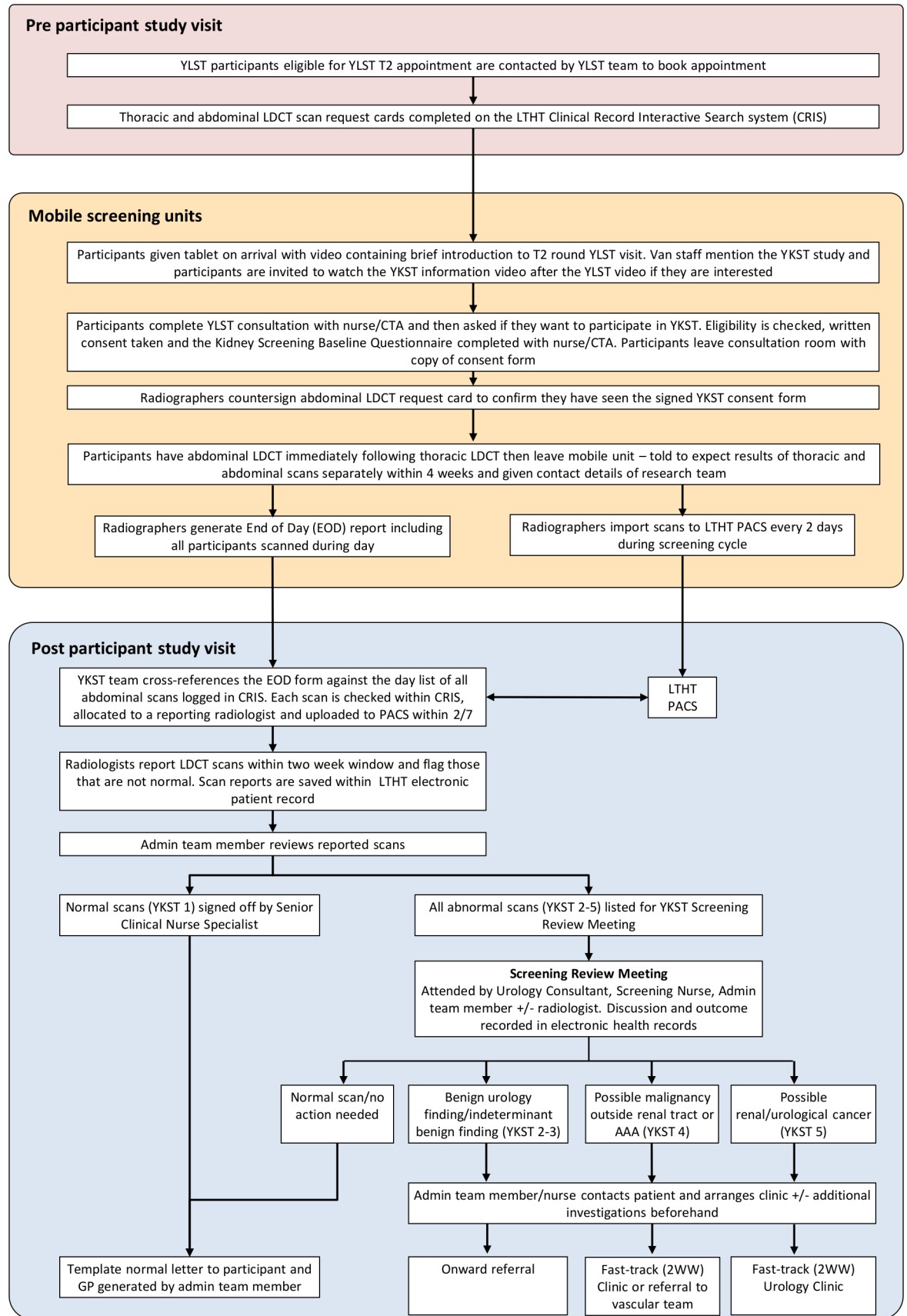

**Figure 2** Main study process map. AAA, abdominal aortic aneurysm; CRIS, Clinical Record Interactive Search System; CTA, clinical trials assistant; EOD, end of day report; LDCT, low-dose CT; LTHT, Leeds Teaching Hospitals Trust; PACS, Picture archiving and communication system; T2, second round of screening within YLST; YKST, Yorkshire Kidney Screening Trial; YLST, Yorkshire Lung Screening Trial.

**Table 1** Outcomes from screening review meeting

| Outcome | Reason | Action | Communication |
|---|---|---|---|
| Normal | No abnormal findings—no action required | Discharge | Patients and GPs sent letter communicating result |
| Benign urological and non-urological findings | Benign findings—no action required | Discharge | Patients and GPs sent letter communicating benign findings and that no further action is required |
| Indeterminate benign finding | Indeterminate finding requiring further elective investigations | Referral to appropriate specialty coordinated by Consultant Urologist and their delegates. Further tests requested as appropriate following recommendations by radiologist. | YKST lead nurse telephones patients (except for adrenal referrals where referral team contacts patients immediately after the referral is made). Patients then contacted by relevant specialty administrative team scheduling appointment and copy sent to GP. |
| Possible malignancy outside the renal tract or AAA | Abnormality requiring immediate further investigation for possible abdominal cancer or AAA | Fast Track 2-week wait appropriate specialty coordinated by Consultant Urologist and their delegates. | YKST lead nurse contacts patients explaining findings, need for further investigations or onward referral. Patients then contacted by relevant specialty administrative team scheduling appointment and copy sent to GP. |
| Possible renal/ urological cancer | Abnormality requiring immediate further investigation for possible renal or urological cancer | During YKST scan review meeting: Consultant Urologist or delegates request contrast scan, refer patient to urology multi-disciplinary team meeting (MDT) and assign them to fast track 2-week wait pathway. | YKST lead nurse telephone patients explaining findings and need for further investigations. Patients then contacted by relevant specialty administrative team scheduling appointment and copy sent to GP. |

AAA, abdominal aortic aneurysm; GP, general practice; YKST, Yorkshire Kidney Screening Trial.

The abdominal CT scans will be classified according to one of the five categories below:
▶ YKST1: normal.
▶ YKST2: benign urological finding.
▶ YKST3: indeterminate benign finding (ie, cholecystitis/pancreatitis).
▶ YKST4: possible malignancy outside renal tract or abdominal aortic aneurysm.
▶ YKST5: possible renal/urological cancer.

Normal scan reports (YKST1) will be reviewed and signed off by a senior clinical nurse specialist. A letter will be sent to the patient and their GP, explaining that their scan was normal and that there are no further actions required.

All scans not reported as normal, as well as any normal LDCT scans with discordant reports after second reads (see quality assurance details below), will be reviewed in a screening review meeting (SRM). SRMs will take place twice weekly and will be attended by a consultant urologist with an interest in renal cancer, a senior clinical nurse specialist and a clinical trials administrator. The administrative team will record the agreed outcome and communicate the results to participants and their GPs according to table 1. Participants will be able to contact the YKST team via the YKST website (www.ykst.org) and the YKST phone number.

## Quality assurance
Ten percent of all normal scans (YKST1) will be selected at random, rereported by a different radiologist and categorised as YKST 1–5 in a second report. The quality of the scans will be assessed both qualitatively using a Likert score from 1 (poor) to 5 (excellent) and quantitatively by selecting a region of interest (ROI). The Likert scale will be recorded by the radiologists for all scans. The ROI assessment will only be performed on the 10% of scans that are second read and will be reported in an addendum to the scan report.

## Qualitative interviews
The interviews will take place over the telephone or video call. The interview schedule will be informed by the Theoretical Framework of Acceptability[18] and explore participants' views on the acceptability of the information provided, the consent process, their thoughts on the combined screening approach and their reasons for accepting or declining the abdominal scan. The interviews will also explore any psychological harm or anxiety that may be experienced by taking part in this combined screening approach. Healthcare professionals who are involved in the study will also be invited to take part in an interview to assess the acceptability of the combined screening approach to staff members. All interviews will be recorded and transcribed.

## Follow-up
To capture the potential downstream harms of the abdominal CT scan, the medical notes of all participants who had an abnormal finding on the abdominal CT will be reviewed 6 months after the scan by the study team; to identify all investigations, procedures, complications, diagnoses and management arising from findings on the abdominal CT. For participants who move out of the study area, all reasonable efforts will be made to determine what their outcome was. Incidental findings will be divided into serious and non-serious based on whether

or not they represent a condition which carries a real prospect of seriously threatening life span, or of having a substantial impact on major body functions or quality of life.[19] The classification of findings will be performed by two clinicians independently based on the clinical information within the patient electronic health records and the list of potentially serious/non-serious incidental findings developed in a previous study for abdominal MRI scans based on consultations with radiologists, review of the literature and the German National Cohort's list of imaging incidental findings[19] and consultation with the clinicians within the research team. Agreement between the two clinicians will be reported by calculating the percentage of findings for which both clinicians agreed on the initial classification. Any discrepancies will be reviewed and discussed at a consensus meeting.

Long-term follow-up will take place between months 20 and 120, and will include the following: number of kidney and other upper abdominal cancers detected and histological subtype, pathological tumour stage and grade; number and details of non-cancer findings; cancer stage at diagnosis; treatments received date and cause of death.

### Psychosocial and other non-physical harms substudy

A subset of approximately 500 participants consisting of all those who have an abnormal CT scan report (YKST 2–5) between March 2022 and October 2022 and a random sample of one-third of those with normal scans (YKST 1) recruited within the same time period will be sent a short questionnaire 3 months and 6 months after the scan to evaluate outcomes in relation to psychological, social and financial harms and dissatisfaction with healthcare. Questionnaires will be sent by post, with participants having the option to complete the questionnaire online and one reminder will be sent 2 weeks after each questionnaire to reduce non-response bias. The questionnaire will include validated measures where possible, including the Psychological Consequences Questionnaire (PCQ),[20] the Short form of the Spielberger State Trait Anxiety Inventory (STAI),[21] the EQ-5D-5L[22] and a single question asking how participants would rate their general health now compared with before they were invited to take part in YKST. The financial consequences of having the scan will be measured using five questions from a previous study[19] and satisfaction with healthcare using the abbreviated measure to assess trust in the medical profession.[23] The questionnaire will also assess participant satisfaction with the information they received and whether they felt they had had sufficient time and information to make the decision whether or not to accept the scan.

### Withdrawal of consent

If participants wish to withdraw from the study, no further data will be collected on them, though we will keep all data collected to that point. All patient withdrawals will be recorded.

### Safety

Adverse events occurring between the time the participants enter the mobile van for their T2 visit and the time that their final result letter is written to them and they are discharged from the study will be recorded and reported in line with Good Clinical Practice.

### Sample size

The maximum sample size is limited to those participants who attend for their YLST T2 visit. Approximately 6500 participants were recruited into YLST and it is estimated that 80% of those will attend for the T2 visit. Recruitment began 2 months into T2 on 10 May 2021 and will run until 31 October 2022. Approximately 4700 individuals will therefore be eligible for inclusion into YKST. If 80%–90% of those take up the additional screening, it will be possible to measure the proportion taking up the additional scan, the primary quantitative outcome of this study, to within 1%. For the qualitative substudy, the principles of information power[24] will be used to decide when to cease data collection but we anticipate interviewing up to 40 participants. We will purposefully sample participants with the aim to include approximately 20 who accept the additional scan and 20 who do not, with a range of ages, sex, ethnicity and socioeconomic status. For the qualitative interviews with healthcare professionals, there are approximately 10 closely involved in the screening process and all will be approached.

### Data analysis
#### Primary outcomes

We will report the proportion of the population attending the T2 screening round who (1) are eligible to take part in YKST; (2) are invited to take part in YKST; (3) consent to the additional scan within YKST and (4) decline taking part in YKST. We will also report these proportions by age, sex, smoking status, ethnicity and socioeconomic status and compare those invited who accept and undergo the abdominal CT scan between demographic subgroups. The additional time required at each stage (obtaining consent, performing, reporting and reviewing the scans and feeding back the results to participants) of the combined screening approach will be reported. Qualitative data evaluating the acceptability of the combined screening approach will be analysed using Framework analysis, guided by the Theoretical Framework of Acceptability.[18] Each transcript will be read by at least two members of the study team with other members of the study team reading some of the transcripts and contributing to discussions about the overall findings.

#### Secondary outcomes

Descriptive summaries of secondary outcomes will be reported. All clinical outcomes will be based on the final diagnosis obtained from the 6-month follow-up data. When reporting the prevalence and stage distribution of RCC, we will present data among the participants who had the abdominal CT as well as among those from the

baseline round of scanning in YLST who either had a renal mass identified in that baseline (T0) lung LDCT or staging investigations for any lung lesions identified and so would have had their full kidneys imaged.

## Patient and public involvement

Two members of the public were involved in the design of this study and contributed to the research proposal prior to submission for funding. They have also commented on all participant facing documentation and continue to contribute to the study as members of the Independent Trial Steering Committee.

## ETHICS AND DISSEMINATION

This study was granted approval by the North West—Preston Research Ethics Committee (reference 21/NW/0021), and the Health Research Authority on 3 February 2021. It has been adopted onto the National Institute for Health Research trial portfolio (reference 290336). The University of Leeds is the sponsor and together with LTHT acts as joint data controller. The trial will have three committees providing oversight: the Trial Management Group (TMG), the Independent Data Monitoring Committee (IDMC) and an Independent Trials Steering Committee (TSC). The TMG will meet on a monthly basis, and will consist of the co-ordinating team based in Cambridge, members of the YKST team based in Leeds as well as the YLST principal investigator, data manager, project manager and lead nurse. The TMG will provide regular monitoring of the trial and provide clinical, scientific and practical advice. The IDMC will meet once or twice a year and will monitor patient safety as well as interim data. The TSC will meet once or twice a year and will provide overall oversight for the trial. The independent members of the IDMC and TSC will include experts in the field of cancer screening, radiology, renal cancer and statistics. The TSC will also include at least one patient/public representative.

Findings from the study will be reported in open-access papers in peer-reviewed journals and presented at national and international conferences. We will also provide a lay summary of the findings on the study website (www.YKST.org).

## DISCUSSION

As the first study of its kind, YKST will assess the feasibility and acceptability of a combined abdominal and lung cancer screening approach and estimate other key uncertainties needed to inform a health economic analysis and future randomised controlled trials. Nesting YKST within an ongoing randomised lung cancer screening trial also provides a unique opportunity to generate the first cohort of participants invited to undergo screening for RCC. Although limited to assessing uptake and acceptability among participants who have already accepted screening for lung cancer and not large enough on its own to enable

precise estimates of prevalence of RCC or an assessment of whether screening for RCC reduces RCC mortality, this cohort will be a valuable foundation for future research.

**Author affiliations**
[1]Department of Public Health and Primary Care, University of Cambridge, Cambridge, UK
[2]Department of Surgery, University of Cambridge, Cambridge, UK
[3]Department of Oncology, University of Cambridge, Cambridge, UK
[4]Department of Radiology, Leeds Teaching Hospitals NHS Trust, Leeds, UK, Leeds, UK
[5]Department of Urology, Leeds Teaching Hospitals NHS Trust, Leeds, UK, Leeds, UK
[6]Division of Infection, Immunity and Respiratory Medicine, Faculty of Biology, Medicine and Health, The University of Manchester, Manchester, UK
[7]Leeds Instituite of Health Sciences, University of Leeds, Leeds, UK
[8]Research and Innovation, Leeds Teaching Hospitals NHS Trust, Leeds, UK
[9]Leeds Diagnosis & Screening Unit, Leeds Institute of Health Sciences, University of Leeds, Leeds, UK
[10]Department of Medical Physics & Engineering, Leeds teaching hospitals NHS Trust, Leeds, UK
[11]Pitcairn Practice, Balmullo Surgery, Fife, UK
[12]College of Medicine and Health, University of Exeter, Exeter, UK
[13]Department of Radiology, University of Cambridge, Cambridge, UK
[14]Department of Radiology, Catholic University Sacro Cuore and Policlinico Universitario Agostino Gemelli, IRCCS, Rome, Italy
[15]Upper Gastro-intestinal and Pancreas Unit, Leeds Teaching Hospitals NHS Trust, Leeds, UK
[16]MRC Epidemiology Unit, University of Cambridge, Cambridge, UK
[17]Leeds Vascular Institute, Leeds Teaching Hospitals NHS Trust, Leeds, UK
[18]Department of Respiratory Medicine, Leeds Teaching Hospitals NHS Trust, Leeds, UK

**Acknowledgements** The authors thank the patient and public representations who have contributed to this study, Phil Alsop and Philip Dondi. They also thank the YLST study team and members of the YKST IDMC (Paul Nathan (Chair), Vicky Goh, Damian Hanbury and Akhtar Nasim) and TSC (Peter Sasieni (Chair), Jonathan Mant, David Nicol, Robert Rintoul, Katie Robb and Jo Waller).

**Contributors** Conceptualisation: GDS and JU-S. Design: GDS, MC, JU-S, SWB, RDN, SR, SHR, ES, TW, AS, GRI, SiB and JC. Draft: AP-R, GDS and JU-S. Revision: SWB, SiB, JC, PAJC, CE, FF, DH, NH, GRI, MK, GM, RDN, SR, SHR, ES, AS, SJS, IS, TW, MW, MC, GDS, JU-S and AP-R.

**Funding** This study is funded by Yorkshire Cancer Research grant number L403C. The qualitative substudy is funded by a grant from Kidney Cancer UK. GDS is supported by The Mark Foundation for Cancer Research, the Cancer Research UK Cambridge Centre (C9685/A25177) and NIHR Cambridge Biomedical Research Centre (BRC-1215-20014). PAJC is supported by the Manchester National Institute for Health Research Manchester Biomedical Research Centre (IS-BRC-1215-20007). The views expressed are those of the author(s) and not necessarily those of the NIHR or the Department of Health and Social Care.

**Competing interests** All authors have completed the Unified Competing Interest form at www.icmje.org/coi_disclosure.pdf (available on request from the corresponding author). GDS has received educational grants from Pfizer, AstraZeneca, and Intuitive. Surgical; consultancy fees from Pfizer, Merck, EUSA Pharma, and CMR Surgical; travel expenses from Pfizer; and speaker fees from Pfizer. All other authors declare that (1) they have no support from or relationships with companies that might have an interest in the submitted work in the previous 3 years; (2) their spouses, partners, or children have no financial relationships that may be relevant to the submitted work; and (3) they have no non-financial interests that may be relevant to the submitted work. The corresponding author has the right to grant on behalf of all authors and does grant on behalf of all authors, an exclusive licence on a worldwide basis to the BMJ Publishing Group Ltd and its Licensees to permit this article (if accepted) to be published in BMJ editions and any other BMJPGL products and sub-licences to exploit all subsidiary rights, as set out in our licence (http://resources.bmj.com/bmj/authors/checklists-forms/licence-for-publication). The corresponding author affirms that the manuscript is an honest, accurate, and transparent account of the study being reported; that no important aspects of the study have been omitted; and that any

discrepancies from the study as planned (and, if relevant, registered) have been explained.

**Patient and public involvement** Patients and/or the public were involved in the design, or conduct, or reporting or dissemination plans of this research. Refer to the Methods section for further details.

**Patient consent for publication** Not required.

**Provenance and peer review** Not commissioned; externally peer reviewed.

**ORCID iDs**
Juliet A Usher-Smith http://orcid.org/0000-0002-8501-2531
Philip A J Crosbie http://orcid.org/0000-0001-8941-4813
Richard D Neal http://orcid.org/0000-0002-3544-2744
Grant D Stewart http://orcid.org/0000-0003-3188-9140

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
