## [Reviewer comments · BMJ Open]

ARTICLE DETAILS

TITLE (PROVISIONAL)	The Yorkshire Kidney Screening Trial (YKST): protocol for a feasibility study of adding non-contrast abdominal CT scanning to screen for kidney cancer and other abdominal pathology within a trial of community-based CT screening for lung cancer
AUTHORS	Usher-Smith, Juliet; Pushpa-Rajah, Angela; Burge, Sarah; Burbidge, Simon; Cartledge, Jon; Crosbie, Philip; Eckert, Claire; Farquhar, Fiona; Hammond, David; Hancock, Neil; Iball, Gareth R; Kimuli, Michael; Masson, Golnessa; Neal, Richard; Rogerson, Suzanne; Rossi, Sabrina; Sala, Evis; Smith, Andrew; Sharp, Stephen; Simmonds, Irene; Wallace, Tom; Ward, Matthew; CALLISTER, Matthew; Stewart, Grant

VERSION 1 – REVIEW

REVIEWER	Chen, Hsin-Yu UCSF
REVIEW RETURNED	26-Apr-2022

GENERAL COMMENTS	Thank you for the invitation to review this draft protocol by Usher-Smith et al., entitled "The Yorkshire Kidney Screening Trial (YKST): protocol for a feasibility study of adding noncontrast abdominal CT scanning to screen for kidney cancer and other abdominal pathology within a trial of community-based CT screening for lung cancer." The protocol described a clinical trial which enrolls participants of a lung cancer LDCT screening study into an additional non-contrast abdominal CT for RCC screening, with the aims to evaluate the uptake and acceptability of such an approach. The design of the trial protocol per se, in terms of recruitment process, choice of endpoints, imaging protocols, statistical power, data collection and analysis are generally reasonable. Therefore, we found this trial could be of scientific interest in a clinical research setting. That being said, we have a few specific concerns, detailed as follows. Major-- 1. The clinical paradigm for RCC is quite different from that of lung cancer. The space of kidney cancer has already been heavily plagued by overdiagnosis and overtreatment primarily associated with the widespread use of cross-sectional imaging [1-2] over the past 30 years. As seen in Fig.1 of Ref[1](linked), the incidence/detection of early-stage RCC increased several fold without improvement in clinical outcomes -- meaning we could be just picking up/treating more low-grade, indolent tumors that are not destined to cause clinical illness or death. Indeed, a significant proportion of the incidentally-found small renal masses on imaging turned out to be either benign pathology (e.g. oncocytoma) or low-grade RCC (who may never die of RCC if left undiagnosed)[3] at
---

partial or radical surgery. These patients can be exposed to harm which far outweighs benefit; there is significant morbidity resulting from surgical intervention, the financial toxicity related to the diagnostic workup, treatment, or sequential follow-up/surveillance imaging, as well as psychosocial impacts. Overall, we felt this was not sufficiently addressed/discussed in the draft protocol.

a. How does this imaging-based screening trial propose to address, or at least not exacerbate the abovementioned issue?

b. Page 14 Line 37: "... video explains ... an estimate that in about 5 out of 1000 eligible people the scan may show evidence of RCC." Given the risk of overdiagnosis and overtreatment, does it make more sense to consult the patients on "numbers need to screen to prevent one RCC-specific death", or "numbers need to screen to prevent one symptomatic RCC presentation", rather than number needed to screen for one diagnosis?

c. Page 11 Line 47: A primary endpoint is the patient uptake from the YLST trial who participate in the YKST trial. We felt that an accurate, unbiased measurement of such endpoint will be contingent on the participants being well-informed not only on the benefit (preventing RCC mortality) and risk (overdiagnosis, overtreatment) of the proposed screening test, but more importantly, the likelihood of each outcome upon which they base their decision. Are the participants consulted with an approximate relative likelihood of preventing RCC mortality vs overdx/overtx, so they can evaluate the risk/benefit and make an informed decision in their best interest?

d. Page 6 Line 13: "Diagnosing RCC at an early stage is therefore central to improving survival."

Page 6 Line 20: "... half of all patients developing the disease [RCC] die from it."

These statements need reference. Is there evidence that early diagnoses improve RCC outcomes, rather than creating lead time and overdiagnosis biases? Regarding mortality, there are estimated 79,000 kidney cancer diagnoses and 13,000 deaths last year in the US[4] - granted there could be a difference among countries/regions.

[1] "Stumbling onto Cancer: Avoiding Overdiagnosis of Renal Cell Carcinoma", Welch HG et al., Am Fam Physician. 2019 Feb 1;99(3):145-147.

<https://www.aafp.org/afp/2019/0201/p145.html>

[2] "The harms of overdiagnosis and overtreatment in patients with small renal masses: a mini-review", Sohlberg EM et al., European urology focus 5.6 (2019): 943-945"

<https://pubmed.ncbi.nlm.nih.gov/30905599/>

[3] "Current Management of Small Renal Masses, Including Patient Selection, Renal Tumor Biopsy, Active Surveillance, and Thermal Ablation, Sanchez A et al., Journal of Clinical Oncology 36.36 (2018): 3591"

<https://ascopubs.org/doi/10.1200/JCO.2018.79.2341>

[4] Cancer Facts and Figures, American Cancer Society

<https://www.cancer.org/content/dam/cancer-org/research/cancer-facts-and-statistics/annual-cancer-facts-and-figures/2022/2022-cancer-facts-and-figures.pdf>

2. We were wondering if this trial would also be a good opportunity to work out the economy, which isn't particularly elaborated in the current draft. We appreciate that it does seek to evaluate the

	subjective perception of financial impact via questionnaires. Nonetheless, we felt that the objective financial aspects will play an essential role in the viability of this approach. How much additional time and effort does it require for the radiologists, specialists and clinical staff involved? If this screening approach is to be adopted for routine clinical practice, how much reimbursement will these efforts translate into (e.g. under the NHS payment system)? In this regard, can the conclusions of this trial be applied or extrapolated to other healthcare systems, such as the US - with predominantly private payers? Or does the study specifically intend to evaluate this approach for the UK? 3. Page 9 Line 35: "As older age and smoking are the two strongest risk factors for RCC[12], this population invited for lung cancer screening are also at higher risk of developing RCC." This seems to suggest the common risk factors of the two are the basis of offering YKST based on YLST's inclusion criteria. As we know smokers are at 15-30x higher risk of developing lung cancer. Similarly, can a hazard ratio of developing RCC be cited for these risk factors? Does the numbers justify using these LC risk factors to select patients for RCC screening, but exclude those without? On a related note, will the participants/invitations be stratified based on their competing comorbidities (e.g. cardiovascular disease, DM) to select for those more likely to benefit from RCC intervention? Why or why not? Also, are genetic risk factors (e.g. VHL syndrome) taken into account? Minor-- 4. Page 10 Line 54-Page 11 Line 3: What's the difference between "acceptability" and "uptake"? Does it mean those who find additional imaging acceptable versus those who actually undergo the scan? In the context of this trial, what would each metric inform? It would benefit the readers to clarify. 5. Page 12 Line 8: One of the primary endpoints is "The additional time required for the combined screening approach." Does this include only the extra time required for the scan? In those with abnormal findings, does it take into account the extra time toward potentially additional scans, clinical workup and management? Why or why not? 6. Page 12 Line 25: "The proportion of participants found to have incidental renal findings on noncontrast CT scanning." Does this mean all incidental findings, or only non-malignant conditions? 7. Page 10 Line 1: "The prevalence of renal masses and RCC found on non-contrast CT screening." What constitutes an RCC finding? Is it solely based off imaging, or does it require a biopsy or nephrectomy tissue diagnosis? This should be clarified. 8. Page 20 Line 30: "kidney ... pathological types". Will the pathological grades of RCC be reported as well? 9. Page 20 Line 40: "Psychosocial and other non-physical harms sub-study." The study plans to conduct questionnaires on ~500 participants to investigate non-physical harms. Among these 500 individuals, how would patients from each category of screening result be represented? (i.e. category 1--unremarkable vs category 5--highly suspicious of RCC, as well as intermediates)
--	---

	Additionally, those who chose to respond to the questionnaire are likely to have a stronger opinion about the trial than those who don't, be it positive or negative. How is this bias corrected? 10. Page 21 Line 11: Withdrawal of consent is defined here. On the other hand, how will the trial handle data from those who does not follow the protocol for any reason, or are lost to follow-up? 11. The other potential confounding factor is the inherent selection bias when the YKST is piggybacked on a YLST trial. Conceivably, an individual who participated in YLST, who (Page 13 Line 50) a. followed smoke cessation and b. attended the 2-year f/u scan per protocol, would be more motivated to take part in a YKST trial, than the scenario where the YKST is offered on the first visit, or to an alternative cohort. Does this trial seek to address the uptake of YKST specifically to the per-protocol participants of YLST, or does it seek to generalize the findings to a different setting (e.g. offering the abdominal scan on the first visit) or a different population? If the latter is the case, how this bias is being corrected, or otherwise accounted for should be discussed.
--	--

REVIEWER	Prabhu, RA Manipal Academy of Higher Education, Nephrology
REVIEW RETURNED	18-May-2022

GENERAL COMMENTS	This is a feasibility and acceptability study of screening for RCC within a cohort which has been screened for lung cancer and which is at high risk for the same. Some questions which need clarification are: Present standard screening method for RCC is ultrasound of the abdomen. As this study involves additional radiation exposure how would this compare with performing a focused screening ultrasound which would be more feasible and acceptable. Since intravenous contrast is not being used what would be the number of lesions which may be missed? Is a cost effectiveness analysis planned. Since abdominal CT done in asymptomatic individuals is associated false positive and incidental findings how do the authors plan to address these. Again sample size is limited by the number of the previous cohort. Since prevalence of RCC is different from lung cancer in this population will this sample have sufficient power to answer the research question. Finally what would be the difference if this study is done in a fresh cohort rather than a two year old already screened one as that would be more representative.
--

VERSION 1 – AUTHOR RESPONSE

Reviewer: 1
Dr. Hsin-Yu Chen, UCSF

Comments to the Author:

Dear Dr. Leeson and the authors,

Thank you for the invitation to review this draft protocol by Usher-Smith et al., entitled "The Yorkshire Kidney Screening Trial (YKST): protocol for a feasibility study of adding noncontrast abdominal CT scanning to screen for kidney cancer and other abdominal pathology within a trial of community-based CT screening for lung cancer."

The protocol described a clinical trial which enrolls participants of a lung cancer LDCT screening study into an additional non-contrast abdominal CT for RCC screening, with the aims to evaluate the uptake and acceptability of such an approach. The design of the trial protocol per se, in terms of recruitment process, choice of endpoints, imaging protocols, statistical power, data collection and analysis are generally reasonable. Therefore, we found this trial could be of scientific interest in a clinical research setting. That being said, we have a few specific concerns, detailed as follows.

Major--

1. The clinical paradigm for RCC is quite different from that of lung cancer. The space of kidney cancer has already been heavily plagued by overdiagnosis and overtreatment primarily associated with the widespread use of cross-sectional imaging [1-2] over the past 30 years. As seen in Fig.1 of Ref[1](linked), the incidence/detection of early-stage RCC increased several fold without improvement in clinical outcomes -- meaning we could be just picking up/treating more low-grade, indolent tumors that are not destined to cause clinical illness or death. Indeed, a significant proportion of the incidentally-found small renal masses on imaging turned out to be either benign pathology (e.g. oncocytoma) or low-grade RCC (who may never die of RCC if left undiagnosed)[3] at partial or radical surgery. These patients can be exposed to harm which far outweighs benefit; there is significant morbidity resulting from surgical intervention, the financial toxicity related to the diagnostic workup, treatment, or sequential follow-up/surveillance imaging, as well as psychosocial impacts. Overall, we felt this was not sufficiently addressed/discussed in the draft protocol.

a. How does this imaging-based screening trial propose to address, or at least not exacerbate the abovementioned issue?

We completely agree with the reviewer that, as with all screening programmes, there are concerns about overdiagnosis and overtreatment within any future kidney cancer screening programme, in particular a screening programme incorporating abdominal CT scanning. Before implementing a programme, there therefore needs to be randomised controlled trial evidence of a reduction in mortality from kidney cancer resulting from screening and demonstration that this reduction in mortality is not outweighed by harms. This study is an important first step in this process and aims to assess the feasibility and acceptability of a combined abdominal and lung cancer screening approach and estimate other key uncertainties needed to inform a health economic analysis and future randomised controlled trials. We have added the following text to the introduction to more explicitly discuss this:

“The fact that RCC incidence is increasing, is largely curable if detected early, and most patients are asymptomatic at the time of diagnosis, has resulted in interest from both the scientific community and patient representatives for the development of an RCC screening programme. In particular, screening and early detection of RCC has been identified as a key research priority in three independent priority setting initiatives over the last five years[3–6]. However, despite three decades of interest in the topic, no definitive studies have been conducted and there remain a number of key uncertainties[8]. These include whether detecting RCC earlier would translate into reductions in mortality or lead to overdiagnosis and overtreatment and whether the benefits at population level would outweigh the potential physical, psychosocial and financial harms. Randomised controlled trials are, therefore, needed. Additionally, despite the increasing incidence, extrapolating from studies in the USA or Japan, the prevalence of RCC among middle-aged adults within the general population in the UK is estimated to be 0.21% (95% CI, 0.14–0.28%)[7]. This means that approximately 500 individuals would need to be screened to identify one person with a RCC unless screening was targeted towards higher-risk individuals[8].”

Additionally, an important component of this feasibility study is to assess potential harms. While the study is not directly able to quantify overdiagnosis and overtreatment, we are collecting data on all

interventions and complications arising from findings on the scans and include a specific sub-study to enable us to evaluate psychological, social and financial harms, and any dissatisfaction with healthcare among participants. The details of that sub-study are already included within the manuscript as below:

“A sub-set of approximately 500 participants consisting of all those who have an abnormal CT scan report (YKST 2-5) between March 2022 and October 2022 and a random sample of one third of those with normal scans (YKST 1) recruited within the same time period will be sent a short questionnaire three months and six months after the scan to evaluate outcomes in relation to psychological, social and financial harms, and dissatisfaction with health care. Questionnaires will be sent by post, with participants having the option to complete the questionnaire online. The questionnaire will include validated measures where possible, including the Psychological Consequences Questionnaire (PCQ)[23], the Short form of the Spielberger State Trait Anxiety Inventory (STAI)[24], the EQ-5D-5L[25], and a single question asking how participants would rate their general health now compared to before they were invited to take part in YKST. The financial consequences of having the scan will be measured using five questions from a previous study[22] and satisfaction with healthcare using the abbreviated measure to assess trust in the medical profession[26].”

To clarify that we will be collecting data on downstream investigations and complications, we have amended the text in the “Follow up” section of the manuscript to read:

“To capture the potential downstream harms of the abdominal CT scan, the medical notes of all participants who had an abnormal finding on the abdominal CT will be reviewed six months after the scan by the study team to identify all investigations, procedures, complications, diagnoses and management arising from findings on the abdominal CT. Findings will be divided into serious and non-serious based on whether or not they represent a condition which carries a real prospect of seriously threatening life span, or of having a substantial impact on major body functions or quality of life.”

b. Page 14 Line 37: " ... video explains ... an estimate that in about 5 out of 1000 eligible people the scan may show evidence of RCC." Given the risk of overdiagnosis and overtreatment, does it make more sense to consult the patients on "numbers need to screen to prevent one RCC-specific death", or "numbers need to screen to prevent one symptomatic RCC presentation", rather than number needed to screen for one diagnosis?

The numbers needed to screen to prevent one RCC-specific death or the numbers needed to screen to prevent one symptomatic RCC presentation are not known. No trials of screening programmes for kidney cancer have yet been conducted. The information provided to participants in this study was therefore based on the best available evidence on the prevalence of RCC. The information sheet did additionally make reference to this uncertainty and the risk of overdiagnosis and overtreatment – “We estimate that in about 5 out of a 1000 people the scan will show evidence of a kidney cancer. These cancers picked up through screening tend to be early and more treatable. However, it has not yet been proven in trials that detecting these cancers by screening reduces deaths – this study is the first step in gathering the evidence to see if this is the case.” And “Occasionally, people may have tests or treatments for findings that were not needed. This is because the finding later turns out to be benign (not a cancer) or is a harmless type of kidney cancer (that would not cause problems even if left alone). It is important you are aware of this possibility before having the scan.” To make this clearer in the manuscript we have amended the text in the methods section as below and included the participant information sheet as a Supplementary File.

“The YKST video explains the context of the study and the benefits and harms of the additional abdominal CT, including an estimate that in about 5 out of 1000 eligible people the scan may show evidence of RCC, the uncertainty over whether detecting cancers in this way reduces deaths from

RCC, the risks associated with the radiation dose and overdiagnosis and the potential to cause anxiety and worry.”

c. Page 11 Line 47: A primary endpoint is the patient uptake from the YLST trial who participate in the YKST trial. We felt that an accurate, unbiased measurement of such endpoint will be contingent on the participants being well-informed not only on the benefit (preventing RCC mortality) and risk (overdiagnosis, overtreatment) of the proposed screening test, but more importantly, the likelihood of each outcome upon which they base their decision. Are the participants consulted with an approximate relative likelihood of preventing RCC mortality vs overdx/overtx, so they can evaluate the risk/benefit and make an informed decision in their best interest?

As we mention in our response to point (b) above, it is not known whether screening for RCC using CT will reduce RCC mortality or the extent to which it will result in overdiagnosis or overtreatment. That is why this feasibility study is important. We were very careful though in the patient information sheet and worked closely with our patient and public representatives to present the potential harms as well as the potential benefits in order to enable participants to make an informed decision. The precise wording is now included within Supplementary File 1. We will be exploring participants' views on the information provided and consent process within the qualitative interviews (the following text is already included within the manuscript: “The interviews will take place over the telephone or video call. The interview schedule will be informed by the Theoretical Framework of Acceptability and explore participants' views on the acceptability of the information provided, the consent process, their thoughts on the combined screening approach, and their reasons for accepting or declining the abdominal scan.”. The psychosocial and other non-physical harms sub-study also includes questions directly asking participants about satisfaction with the information they received and whether they felt they had sufficient time and information to make the decision to have the abdominal scan. We will, therefore, be able to report on whether participants felt the information provided was sufficient both qualitatively and quantitatively. We have added the following text to the manuscript where we describe the questions included within the questionnaire:

“The questionnaire will also assess participant satisfaction with the information they received and whether they felt they had had sufficient time and information to make the decision whether or not to accept the scan.”

d. Page 6 Line 13: "Diagnosing RCC at an early stage is therefore central to improving survival."

Page 6 Line 20: "... half of all patients developing the disease [RCC] die from it."

These statements need reference. Is there evidence that early diagnoses improve RCC outcomes, rather than creating lead time and overdiagnosis biases? Regarding mortality, there are estimated 79,000 kidney cancer diagnoses and 13,000 deaths last year in the US[4] - granted there could be a difference among countries/regions.

[1]"Stumbling onto Cancer: Avoiding Overdiagnosis of Renal Cell Carcinoma", Welch HG et al., Am Fam Physician. 2019 Feb 1;99(3):145-147.

<https://www.aafp.org/afp/2019/0201/p145.html>

[2] "The harms of overdiagnosis and overtreatment in patients with small renal masses: a mini-review", Sohlberg EM et al., European urology focus 5.6 (2019): 943-945"

<https://pubmed.ncbi.nlm.nih.gov/30905599/>

[3] "Current Management of Small Renal Masses, Including Patient Selection, Renal Tumor Biopsy, Active Surveillance, and Thermal Ablation, Sanchez A et al., Journal of Clinical Oncology 36.36 (2018): 3591"

<https://ascopubs.org/doi/10.1200/JCO.2018.79.2341>

[4] Cancer Facts and Figures, American Cancer Society <https://www.cancer.org/content/dam/cancer-org/research/cancer-facts-and-statistics/annual-cancer-facts-and-figures/2022/2022-cancer-facts-and-figures.pdf>

We have referenced these statements as follows. "Diagnosing RCC at an early stage is therefore central to improving survival." Bindi et al (<https://pubmed.ncbi.nlm.nih.gov/28412331/>) demonstrate clear evidence of a 97% CSS at 10 year for stage 1a tumours, 87% for stage 1b, 78% for stage 2a and 66% for stage 2b+. Showing that detection of the cancer at an earlier stage will enable management resulting in greater 10y CSS. "... half of all patients developing the disease [RCC] die from it." This data comes from a UK data source (<https://www.cancerresearchuk.org/health-professional/cancer-statistics/statistics-by-cancer-type/kidney-cancer>), as this is a UK trial.

2. We were wondering if this trial would also be a good opportunity to work out the economy, which isn't particularly elaborated in the current draft. We appreciate that it does seek to evaluate the subjective perception of financial impact via questionnaires. Nonetheless, we felt that the objective financial aspects will play an essential role in the viability of this approach. How much additional time and effort does it require for the radiologists, specialists and clinical staff involved? If this screening approach is to be adopted for routine clinical practice, how much reimbursement will these efforts translate into (e.g. under the NHS payment system)? In this regard, can the conclusions of this trial be applied or extrapolated to other healthcare systems, such as the US - with predominantly private payers? Or does the study specifically intend to evaluate this approach for the UK?

We agree with the reviewer that a health economic analysis of the costs and benefits of this approach is absolutely essential. We will be collecting all the data required for that analysis within this feasibility study, including the time taken for each stage of the screening pathway and all downstream investigations, diagnoses and treatments. These outcomes are all already detailed within the manuscript, with the additional time mentioned specifically by the reviewer being one of the primary outcome measures. We are then planning a health economic analysis which will follow on from this feasibility study. That is a substantial separate piece of work though and describing that in full within this protocol for the feasibility study is not possible. We have added reference to that subsequent piece of work though and how this feasibility study will feed into it at the following points in the manuscript:

Introduction: "Nested within YLST, the Yorkshire Kidney Screening Trial (YKST) will take advantage of this unique opportunity to assess the feasibility and acceptability of offering an additional non-contrast abdominal CT at the same time as the thoracic LDCT as a combined abdominal and lung cancer screening approach and to estimate other key uncertainties needed to inform a health economic analysis and subsequent randomised controlled trials within future screening programmes."

Discussion: "As the first study of its kind, YKST will assess the feasibility and acceptability of a combined abdominal and lung cancer screening approach and estimate other key uncertainties needed to inform a health economic analysis and future randomised controlled trials."

3. Page 9 Line 35: "As older age and smoking are the two strongest risk factors for RCC[12], this population invited for lung cancer screening are also at higher risk of developing RCC." This seems to suggest the common risk factors of the two are the basis of offering YKST based on YLST's inclusion criteria. As we know smokers are at 15-30x higher risk of developing lung cancer. Similarly, can a hazard ratio of developing RCC be cited for these risk factors? Does the numbers justify using these LC risk factors to select patients for RCC screening, but exclude those without?

Over 95% of RCC deaths in the UK arise in individuals over 50 years and the relative risk for RCC compared with never smokers is 1.35 for current smokers and 1.22 for ex-smokers. The justification for using these lung cancer risk factors to select patients for RCC screening is based both on the strength of age and smoking status as risk factors but also on the efficiency that would be gained from combining the abdominal and lung CT scans. We have included those numbers and made our justification clearer in the text as below:

“The gold standard test for detecting and investigating renal masses is a contrast-enhanced abdominal computed tomography (CT) scan. It is not feasible to use a contrast-enhanced CT as a stand-alone screening test for RCC due to the relatively high radiation dose and cost, particularly given the low prevalence of RCC. However, using a non-contrast CT scan and combining that with the thoracic low-dose CT (LDCT) scans recommended in the USA in adults aged 50-80 years who have a 20 pack-year smoking history or have quit smoking within the past 15 years [11] and currently being reviewed by the UK National Screening Committee has been proposed[8]. Over 95% of deaths from RCC in the UK occur in those aged over 50 and the relative risk for RCC compared with never smokers is 1.35 for current smokers and 1.22 for ex-smokers. This combined approach would therefore reduce both the costs and radiation, while also targeting those at higher risk due to their age and smoking status.”

On a related note, will the participants/invitations be stratified based on their competing comorbidities (e.g. cardiovascular disease, DM) to select for those more likely to benefit from RCC intervention? Why or why not?

Also, are genetic risk factors (e.g. VHL syndrome) taken into account?

As described in the participants and recruitment section of the manuscript and in Figure 1, all those who participants within YLST who are invited back to the YLST T2 visit are invited to participate in YKST unless they have had an abdominal or thoracic CT within the last 6 months, are unable to have an abdominal or thoracic LCDT, have a previous diagnosis of RCC or are unable to provide informed consent. No additional participant characteristics or genetic risk factors are used to determine eligibility. We will be collecting data on additional risk factors of RCC (whether they have a diagnosis of diabetes or hypertension, whether they take antihypertensive medication, if they have a family history of kidney or pancreatic cancer and their average weekly alcohol consumption – included within the manuscript in the “Invitation process, consent and baseline data collection” section but these are not used to determine eligibility. This is so that the approach in this feasibility reflects how a possible combined RCC and lung screening might be implemented in the future, where such information would not necessarily be known in advance.

Minor--

4. Page 10 Line 54-Page 11 Line 3: What's the difference between "acceptability" and "uptake"? Does it mean those who find additional imaging acceptable versus those who actually undergo the scan? In the context of this trial, what would each metric inform? It would benefit the readers to clarify.

Uptake is the proportion of individuals invited to have an additional abdominal CT while attending a second round of lung cancer screening who take up the offer of the abdominal CT (defined within Primary outcome measure 1). Acceptability is a multi-component measure reflecting the views of those who chose to take up the additional scan and will inform the design of any future trials. The section on “Qualitative interviews” describes how we will assess acceptability using the Theoretical Framework of Acceptability and how we will explore participants’ views on the acceptability of the information provided, the consent process, their thoughts on the combined screening approach, and their reasons for accepting or declining the abdominal scan.

5. Page 12 Line 8: One of the primary endpoints is "The additional time required for the combined screening approach." Does this include only the extra time required for the scan? In those with abnormal findings, does it take into account the extra time toward potentially additional scans, clinical workup and management? Why or why not?

The additional time required for the combined screening approach includes the extra time required within the screening programme up until the point at which the participant is either sent a normal letter or referred on for further care. This includes the additional time taken for the scan as well the time taken for consent and reporting. We have clarified this in the text:

"The additional time required at each stage (obtaining consent, performing, reporting and reviewing the scans, and feeding back the results to participants) of the combined screening approach will be reported."

As already described within the manuscript at the end of the "Outcome measures" section, we will separately be collecting data on "further investigations, procedures and management of findings identified on abdominal CT to estimate the individual and health system burden of incidental findings and on the agreement of radiologists reporting the scans and the abdominal CT scan dose and quality to assess the safety."

6. Page 12 Line 25: "The proportion of participants found to have incidental renal findings on noncontrast CT scanning." Does this mean all incidental findings, or only non-malignant conditions?

This means all renal findings that are not a renal mass or RCC.

7. Page 10 Line 1: "The prevalence of renal masses and RCC found on non-contrast CT screening." What constitutes an RCC finding? Is it solely based off imaging, or does it require a biopsy or nephrectomy tissue diagnosis? This should be clarified.

All the clinical outcomes will be based on the final diagnoses. These will be obtained at the six month follow-up review. This has been clarified in the "Data analysis" section of the manuscript as below:

"Descriptive summaries of secondary outcomes will be reported. All clinical outcomes will be based on the final diagnosis obtained from the six month follow-up data. When reporting the prevalence and stage distribution of RCC, we will present data among the participants who had the abdominal CT as well as among those from the baseline round of scanning in YLST who either had a renal mass identified in that baseline (T0) lung LDCT or staging investigations for any lung lesions identified and so would have had their full kidneys imaged."

8. Page 20 Line 30: "kidney ... pathological types". Will the pathological grades of RCC be reported as well?

Pathological grade of RCC will be reported. We have amended that sentence to read:

"Long term follow-up will take place between months 20 and 120, and will include: number of kidney and other upper abdominal cancers detected and histological subtype, pathological tumour stage and grade; number and details of non-cancer findings; cancer stage at diagnosis; treatments received; date and cause of death."

9. Page 20 Line 40: "Psychosocial and other non-physical harms sub-study." The study plans to conduct questionnaires on ~500 participants to investigate non-physical harms. Among these 500 individuals, how would patients from each category of screening result be represented? (i.e. category

1--unremarkable vs category 5--highly suspicious of RCC, as well as intermediates) Additionally, those who chose to respond to the questionnaire are likely to have a stronger opinion about the trial than those who don't, be it positive or negative. How is this bias corrected?

The 500 individuals within the sub-study will consist of all those who have an abnormal CT report (YKST 2-5) between March 2022 and October 2022 and a random sample of one third of those with normal scans (YKST 1) recruited within the same time period. We have clarified that within the manuscript as below:

“A sub-set of approximately 500 participants consisting of all those who have an abnormal CT scan report (YKST 2-5) between March 2022 and October 2022 and a random sample of one third of those with normal scans (YKST 1) recruited within the same time period will be sent a short questionnaire three months and six months after the scan to evaluate outcomes in relation to psychological, social and financial harms, and dissatisfaction with health care.”

As the reviewer suggests, it is possible that those who respond to the questionnaire may have a stronger opinion about the trial than those who don't. This is a limitation of all questionnaire studies as a result of response bias. We have designed this study to limit this bias by maximising the response rate. This includes working closely with our patient and public representatives to make the questionnaire items simple and clear to respond to and the questionnaires as short as possible, and offering the option to complete the questionnaire either on paper or electronically. We also plan to send reminder questionnaires after two weeks. We have updated the section describing that study to reflect that as below:

“Questionnaires will be sent by post, with participants having the option to complete the questionnaire online and one reminder will be sent two weeks after each questionnaire to reduce non-response bias.”

We do not plan to attempt to 'correct' any remaining bias in the analysis. Non-response weighting does not eliminate bias as it assumes that those participants who do not respond within a given group have the same opinions as all others in that group. Instead, we will describe the characteristics of those who respond and those who have not responded and will discuss in detail the extent of non-response bias and the potential effect this has on the findings when we report that component of the study.

10. Page 21 Line 11: Withdrawal of consent is defined here. On the other hand, how will the trial handle data from those who does not follow the protocol for any reason, or are lost to follow-up?

In terms of participants who do not follow the protocol, this will mainly occur where individuals consent to YKST but then do not have the CT scan. After this point their active involvement in the study ends and no further non-compliance is possible. To ensure assessment of the key primary objective of 'quantifying the uptake of non-contrast abdominal CT to screen for RCC and other abdominal pathology as part of a combined screening modality with thoracic LDCT within a lung health check' in the way that the study was set up to demonstrate, we will measure the proportion of those invited who consent and then actually have the scan. We will define 'uptake' as actually having the scan and then report the number of people who thought they wanted it (i.e. consented) but then couldn't go ahead. For participants who move out of the study area, all reasonable efforts will be made to determine what their outcome was by enquiring with general practitioner and any other hospitals treatment was undertaken in. We have added the following text to the 'Follow-up' section:

For participants who move out of the study area, all reasonable efforts will be made to determine what their outcome was.

We have amended the data analysis for primary outcomes section to:

We will also report these proportions by age, sex, smoking status, ethnicity and socio-economic status, and compare those invited who accept and undergo the abdominal CT scan between demographic subgroups.

11. The other potential confounding factor is the inherent selection bias when the YKST is piggybacked on a YLST trial. Conceivably, an individual who participated in YLST, who (Page 13 Line 50) a. followed smoke cessation and b. attended the 2-year f/u scan per protocol, would be more motivated to take part in a YKST trial, than the scenario where the YKST is offered on the first visit, or to an alternative cohort. Does this trial seek to address the uptake of YKST specifically to the per-protocol participants of YLST, or does it seek to generalize the findings to a different setting (e.g. offering the abdominal scan on the first visit) or a different population? If the latter is the case, how this bias is being corrected, or otherwise accounted for should be discussed.

As the reviewer highlights, this feasibility study will only be able to assess the uptake of YKST within the specific cohort of participants within YLST who have attended for the T2 visit. We will not seek specifically to adjust for bias to enable generalisation to a different setting and will not be able to assess whether uptake would be different if offered on the first visit or it offering the additional CT before the first visit might increase uptake to lung cancer screening. This is a limitation of the study and, if this study shows that the combined screening approach is feasible, would be the aim of a subsequent study. We have included reference to this in the discussion section as below:

“Although limited to assessing uptake and acceptability among participants who have already accepted screening for lung cancer and not large enough on its own to enable precise estimates of prevalence of RCC or an assessment of whether screening for RCC reduces RCC mortality, this cohort will be a valuable foundation for future research.”

Best regards,

Reviewer: 2

Dr. RA Prabhu, Manipal Academy of Higher Education

Comments to the Author:

This is a feasibility and acceptability study of screening for RCC within a cohort which has been screened for lung cancer and which is at high risk for the same. Some questions which need clarification are:

Present standard screening method for RCC is ultrasound of the abdomen. As this study involves additional radiation exposure how would this compare with performing a focused screening ultrasound which would be more feasible and acceptable.

We are not aware of any organised national screening programmes for renal cancer using ultrasound or any other imaging modality. Ultrasound is an attractive proposition for screening as it is non-invasive, cheap and does not involve radiation. We are unconvinced about feasibility of USS. We have previously evaluated bolting-on an ultrasound of the kidneys to the abdominal aortic aneurysm screening programme in the UK, but a major challenge is training the technicians to learn this

procedure and identify lesions of variable sizes in often obese patients. Additionally, we have data to suggest that members of the public prefer the concept of combined CT chest and abdomen to USS as a screening tool (<https://pubmed.ncbi.nlm.nih.gov/33115457/>).

Since intravenous contrast is not being used what would be the number of lesions which may be missed?

The evidence base is scanty on this point. However, in small cohorts of patients having non-contrast and then contrast enhanced CTs, the AUC for identification of T1a renal masses varies from 0.75-0.92 (<https://journals.sagepub.com/doi/full/10.1177/2058460119849706>). It is not possible to state the number of lesions missed, but this will be estimated at the end of the study.

Is a cost effectiveness analysis planned.

Yes, we are planning a health economic analysis which will follow on from this feasibility study. That is a substantial separate piece of work though and describing that in full within this protocol for the feasibility study is not possible. We have added reference to that subsequent piece of work though and how this feasibility study will feed into it at the following points in the manuscript:

Introduction: "Nested within YLST, the Yorkshire Kidney Screening Trial (YKST) will take advantage of this unique opportunity to assess the feasibility and acceptability of offering an additional non-contrast abdominal CT at the same time as the thoracic LDCT as a combined abdominal and lung cancer screening approach and to estimate other key uncertainties needed to inform a health economic analysis and subsequent randomised controlled trials within future screening programmes."

Discussion: "As the first study of its kind, YKST will assess the feasibility and acceptability of a combined abdominal and lung cancer screening approach and estimate other key uncertainties needed to inform a health economic analysis and future randomised controlled trials."

Since abdominal CT done in asymptomatic individuals is associated false positive and incidental findings how do the authors plan to address these.

Assessing the potential harms of the abdominal CT with respect to false positive and incidental findings is a key component of this feasibility study. As we describe in our response above to Reviewer 1, while the study is not directly able to quantify overdiagnosis and overtreatment, we are collecting data on all interventions and complications arising from findings on the scans and include a specific sub-study to enable us to evaluate psychological, social and financial harms, and any dissatisfaction with healthcare among participants. The details of that sub-study are already included within the manuscript as below:

"A sub-set of approximately 500 participants consisting of all those who have an abnormal CT scan report (YKST 2-5) between March 2022 and October 2022 and a random sample of one third of those with normal scans (YKST 1) recruited within the same time period will be sent a short questionnaire three months and six months after the scan to evaluate outcomes in relation to psychological, social and financial harms, and dissatisfaction with health care. Questionnaires will be sent by post, with participants having the option to complete the questionnaire online. The questionnaire will include validated measures where possible, including the Psychological Consequences Questionnaire (PCQ)[23], the Short form of the Spielberger State Trait Anxiety Inventory (STAI)[24], the EQ-5D-5L[25], and a single question asking how participants would rate their general health now compared to before they were invited to take part in YKST. The financial consequences of having the scan will be measured using five questions from a previous study[22] and satisfaction with healthcare using the abbreviated measure to assess trust in the medical profession[26]."

To clarify that we will be collecting data on downstream investigations and complications, we have amended the text in the “Follow up” section of the manuscript to read:

“To capture the potential downstream harms of the abdominal CT scan, the medical notes of all participants who had an abnormal finding on the abdominal CT will be reviewed six months after the scan by the study team to identify all investigations, procedures, complications, diagnoses and management arising from findings on the abdominal CT. Findings will be divided into serious and non-serious based on whether or not they represent a condition which carries a real prospect of seriously threatening life span, or of having a substantial impact on major body functions or quality of life.”

Again sample size is limited by the number of the previous cohort. Since prevalence of RCC is different from lung cancer in this population will this sample have sufficient power to answer the research question.

The primary outcomes of this study are

1. The proportion of individuals invited to have an additional abdominal CT while attending a second round of lung cancer screening who take up the offer of the abdominal CT;
2. The acceptability to participants of combined lung and RCC screening by non-contrast CT scanning;
3. The acceptability to healthcare professionals involved in the combined screening approach; and
4. The additional time required for the combined screening approach.

As described in the sample size section of the manuscript, we will have sufficient power to estimate the proportion of individuals invited who take up the offer to within 1%. The other primary outcomes will be assessed using qualitative data, requiring only approximately 40 participants and 10 health care professionals.

The study will not have sufficient power to enable precise estimates of the secondary outcomes. However, these are not the primary aims of the study. If this study shows that combining abdominal CT scans with lung LDCT scans within lung cancer screening programmes is feasible, then further, larger, studies would be required to assess those outcomes. We have added reference to this into the Discussion section:

“Although limited to assessing uptake and acceptability among participants who have already accepted screening for lung cancer and not large enough on its own to enable precise estimates of prevalence of RCC or an assessment of whether screening for RCC reduces RCC mortality, this cohort will be a valuable foundation for future research..”

Finally, what would be the difference if this study is done in a fresh cohort rather than a two year old already screened one as that would be more representative.

We do not know what the difference would be if this study were conducted on a different cohort. That is a limitation of this study and will be discussed in detail in the manuscript in which we report the results of the study. We have included reference to this in the Discussion section of the manuscript as below:

“Although limited to assessing uptake and acceptability among participants who have already accepted screening for lung cancer and not large enough on its own to enable precise estimates of

prevalence of RCC or an assessment of whether screening for RCC reduces RCC mortality, this cohort will be a valuable foundation for future research.”

VERSION 2 – REVIEW

REVIEWER	Chen, Hsin-Yu UCSF
REVIEW RETURNED	05-Aug-2022

GENERAL COMMENTS	Thank you for this careful revision. We have no additional questions/comments at this time.
---

REVIEWER	Prabhu, RA Manipal Academy of Higher Education, Nephrology
REVIEW RETURNED	08-Aug-2022

GENERAL COMMENTS	Authors have addressed all the comments raised
--